# Evaluating Opt-In Vaginal Human Papillomavirus Self-Sampling: Participation Rates and Detection of High-Grade Lesions (CIN2+) among Unscreened Japanese Women Aged 30–39

**DOI:** 10.3390/healthcare12050599

**Published:** 2024-03-06

**Authors:** Ito Taro, Toshimichi Onuma, Tetsuji Kurokawa, Yoko Chino, Akiko Shinagawa, Yoshio Yoshida

**Affiliations:** 1Department of Obstetrics and Gynecology, Red Cross Fukui Hospital, Fukui 918-8501, Japan; taroito@u-fukui.ac.jp; 2Department of Obstetrics and Gynecology, Faculty of Medical Sciences, University of Fukui, Fukui 910-1193, Japan; toonuma@u-fukui.ac.jp (T.O.); sngw@u-fukui.ac.jp (A.S.); 3Department of Obstetrics and Gynecology, Fukui-Ken Saiseikai Hospital, Fukui 918-8503, Japan; kurotetu@u-fukui.ac.jp; 4Department of Obstetrics and Gynecology, Tannan Regional Medical Center, Fukui 916-8515, Japan; yoyoyo@u-fukui.ac.jp

**Keywords:** cervical cancer, HPV self-sampling, opt-in discipline, PCR-based HPV test, screening, unscreened women

## Abstract

Cervical cancer incidence is increasing among Japanese women, which is partly attributed to low screening rates. This study examined the implementation of opt-in human papillomavirus (HPV) self-sampling among Japanese women aged 30–39 years who had not undergone cervical cancer screening, focusing on those requiring preconception care. The responses to the opt-in approach and effectiveness in detecting cervical squamous intraepithelial neoplasia 2 or worse (CIN2+) were evaluated. Participants used the Evalyn^®^ Brush for self-sampling, with HPV testing conducted using the Cobas 4800 system (version 2.2.0). Out of 3489 eligible, unscreened women from four municipalities in Fukui Prefecture, only 10.6% (370/3489) requested the self-sampling kit. Of these, 77.3% (286/370) returned the kit (HPV testing rate: 8.2% (286/3489)). The HPV positivity rate was 13.7% (39/285), yet only 61.5% (24/39) of those with positive HPV results proceeded to cytology testing. Subsequently, three cases of CIN2+ were detected (10.5/1000). While this study demonstrated a reasonable kit return rate and indicated the capability of opt-in HPV self-sampling to detect CIN2+ cases in unscreened women, the low ordering rate of kits and suboptimal compliance for follow-up cytology testing highlight significant challenges. The findings suggest the need for more effective strategies to enhance participation in cervical cancer screening programs.

## 1. Introduction

The incidence of cervical cancer among women of reproductive age is increasing in Japan. Data from the 2020 Global Cancer Observatory revealed that the incidence of cervical cancer in Japan among individuals in the age group of 20–44 years was 27.7 per 100,000 people, which was significantly higher than that in Australia (10 per 100,000 people) and the United States (9 per 100,000 people) [1]. This alarming trend has highlighted the urgent need to implement measures to reduce the incidence of cervical cancer in reproductive-age people in Japan. One significant contributing factor to the high incidence and mortality rates of cervical cancer in Japan is the low screening rate [1,2]. The cervical cancer screening rate stands at 42.1% in Japan, which is markedly lower than that in the United States (84.5%) and the United Kingdom (78.1%) [3]. This emphasizes the need to encourage women, especially those who have not participated in screening programs, to undergo cervical cancer screening. Moreover, the average age of women at first childbirth in Japan has increased from 28 years in 2000 to 30.7 years in 2019 [4], marking a demographic shift that aligns with the finding that advanced cervical cancers in most cases occur in women who have not undergone screening [5]. Public subsidies for the vaccination of 7th–10th-grade girls were introduced in November 2010. However, in 2020, women aged 30–39 years were not eligible for the HPV vaccine program when they were 12–16 years old [6]. Among women aged 30–39 years, the main reason reported for not participating in cervical cancer screening was the lack of opportunity to undergo screening [7]. Therefore, as part of routine preconception care, cervical cancer screening programs should be improved and encouraged among women aged 30–39 years [8].

Persistent infections with human papillomaviruses (HPV) have been recognized as a cause of cervical cancer [9]. Compared with cytology testing alone, HPV tests are known for their higher sensitivity in detecting cervical squamous intraepithelial neoplasia 2 or worse (CIN2+) [9,10,11]. The process of HPV self-sampling for cervical cancer detection involves participants collecting a vaginal sample themselves and sending it for HPV testing [12,13]. The concordance rate between HPV self-sampling and physician-collected sampling is approximately 85%–90%. Moreover, the sensitivity of HPV self-sampling for detecting CIN2+ is comparable to that of the sampling conducted by physicians [12,14,15]. In a previous study, no significant difference was noted in the prevalence of CIN2+ per 1000 screened participants between HPV self-sampling and sampling by medical personnel [16]. The use of HPV self-sampling strategies is associated with many advantages, including convenience, reduced costs, the ability to obtain a sample in the office or at home, avoiding the hassles associated with pelvic examinations, and social and cultural acceptance [17]. In a recent meta-analysis, a comparison of self-sampling procedures with clinician-conducted sampling revealed nearly double the probability of cancer screening uptake (relative risk: 1.8; 95% confidence interval: 1.7–2.0) [18]. During the SARS-CoV-2 pandemic, HPV self-sampling contributed significantly to improved cervical cancer screening uptake in Sweden [19]. Notably, HPV self-sampling has not yet been implemented as a screening method in Japan [20]. Importantly, previous studies have reported that self-sampling is an effective strategy for motivating individuals who have not participated in cervical cancer screening previously, with the effectiveness of self-sampling surpassing that of conventional mail-out invitations [21,22,23,24,25,26].

Unscreened women can be provided with HPV self-sampling kits using either opt-in or opt-out approaches. The opt-out method involves sending HPV self-sampling kits to unscreened women without requiring their prior consent, whereas the opt-in method sends kits only after confirming their willingness to receive them. Studies have shown that the opt-out approach is more effective than the opt-in method in terms of encouraging the participation of unscreened women in cervical cancer screening [21]. However, the opt-out strategy has drawbacks in that it is costlier and less environmentally sustainable. For instance, approximately 10–20% of the kits in opt-in methods are not utilized, but this figure increases to approximately 70–80% in opt-out methods, as shown in studies from Sweden and Slovenia [26,27]. Consequently, the kit return rate, which serves as an indicator of compliance, is higher with the opt-in approach than with the opt-out method [26,27]. Some studies have reported that opt-out screening is more cost-effective than opt-in screening [28,29]. In a report from Norway, which uses the opt-out HPV self-sampling method, when estimating the average cost of CIN2+ detection, the cost per CIN2+ detected was the lowest in the opt-out group, followed by a 2% higher cost in the opt-in group. This study concluded that the preferred HPV self-sampling approach is ultimately left up to the decision makers and their willingness to pay for these health benefits [30].

The Netherlands has adopted opt-in HPV self-sampling as a strategy to encourage unscreened women to participate in cervical cancer screening [13]. However, the compliance among unscreened women in Japan toward the opt-in method remains unexplored. Therefore, this study was designed to assess compliance with opt-in HPV self-sampling within the framework of actual cervical cancer screening among unscreened Japanese women aged 30–39 years. Our objectives were two-fold: first, to ascertain the response rate for opt-in HPV self-sampling among these unscreened women, and second, to determine the detection rate of CIN2+ among these unscreened women using opt-in HPV self-sampling.

## 2. Materials and Methods

### 2.1. Study Population

For women aged >20 years, cytology screening for cervical cancer is recommended every 2 years in Japan [31]. Organized cervical cancer screening is mainly conducted in municipalities [32]. Screening invitation and results were mailed to participants. Those who have not undergone organized cervical cancer screening are recorded in the municipality. Previous studies have involved various unscreened populations, such as women who have not been screened in over 10 years or those who did not undergo cervical cancer screening after a reminder was sent [26,33]. Compared with the current reminder letter policy, HPV self-sampling could be more effective for women who have been unscreened for 5 or 10 years [34]. An unscreened term of 5 years or longer was set to select eligible participants in this study. Fukui Prefecture had the same number of hospitals in a 2021 survey compared to other prefectures in Japan (Fukui, 8.8 per 100,000 population; Japanese average, 6.5 per 100,000 population) [35]. The incidence of cervical cancer in Fukui Prefecture was 16.5 per 100,000 population, which did not differ from the national average of 16.8 per 100,000 population in 2019 in Japan [36]. The total population of Echizen City, Katsuyama City, Ono City, and Takahama Town in 2020 was 144,373, of which 6991 were women aged 30–39 [37]. Eligible participants in the present study included women aged 30–39 years; who were registered as residents in Echizen City, Katsuyama City, Ono City, and Takahama Town in Fukui Prefecture in Japan; who had not undergone cervical cancer screening in at least 5 years; and who provided written informed consent. Pregnant women were excluded at the time of the opt-in consent because they had already undergone cytology testing during their prenatal checkup. Since the 30s are the central age of pregnancy in Japan, this study included women who became pregnant after returning the self-sampling kit.

### 2.2. Study Setting

We analyzed compliance and CIN2+ detection for opt-in HPV self-sampling among unscreened women in Japan. The Evalyn^®^ Brush (Rovers Medical Devices, Lekstraat, The Netherlands) was used as a self-sampling device in this study, given its excellent test agreement rate with HPV physician-sampling and same CIN2+ detection rate as that of HPV physician-sampling [15,38,39]. Importantly, the HPV DNA test results of the samples obtained using the Evalyn Brush remained unaffected by the sample storage conditions [40]. The Evalyn Brush comes with an instruction manual created by the Japan Cancer Society. Participants used this as a reference when conducting HPV self-sampling. Self-sampling was performed by participants using this manual in person, as described in a previous study [15].

In May 2020, unscreened women from four municipalities were recruited. We sent a letter to unscreened women to confirm their wish to perform HPV self-sampling and included a cervical cancer screening status questionnaire. The deadline was set as 31 July 2020. The returned questionnaire was assessed to determine if the respondents were eligible to take part in this study. In early August 2020, the HPV self-sampling device and a consent form describing eligibility to this study were sent to unscreened women who wished to take part in the study. Participants placed the self-sampling device in an envelope and then sent the sampling kit and consent form to the University of Fukui by 31 August 2020. Thereafter, samples obtained using the Evalyn Brush were tested by the Fukui Prefecture Health Care Association.

In this study, we used the Cobas 4800, a polymerase chain reaction (PCR)-based HPV DNA test, which detects CIN2+ at a higher rate than the signal-based test [28,41]. In addition, we have shown that the combination of an Evalyn Brush and Cobas 4800 has strong concordance rates with HPV physician-sampling and high sensitivity for CIN2+ detection in Japanese women [15]. We measured HPV16, HPV18, and 12 other high-risk (hr) HPV genotypes (i.e., -31, -33, -35, -39, -45, -51, -52, -56, -58, -59, -66, and -68) on Cobas 4800 using a previously reported method and the manufacturer’s protocol [15,42]. The Cobas 4800 System Software (version 2.2.0) was used to perform all assays, run validations, and automatically display reportable HPV test results. For each HPV type, results were provided as positive, negative, or invalid. A failed result is generated when no results are available due to incomplete PCR tests. Invalid and failed results were defined as unmeasurable in this study.

HPV test results from self-sampling were communicated to participants via mail. In cases of HPV-positive results, participants were instructed to undergo organized cervical cancer screening by cytology in Fukui Prefecture. In cases of HPV-negative results, organized cervical cancer screening by cytology was advised. The cytology screening procedure was the same for both groups. Cytology results were mailed to participants. If the cytology result indicated atypical squamous cells of undetermined significance (ASCUS) or worse, participants were instructed to go to a clinic or hospital in Fukui Prefecture. Based on the Japanese Obstetrics and Gynecology Practice Guidelines 2020 Edition, if the cytology result indicated a low-grade squamous intraepithelial lesion (LSIL) or worse or ASCUS with high-risk HPV positivity, biopsy by colposcopy was performed [31]. All colposcopy results were sent to the Fukui Prefectural Health Care Association. We investigated the cytology and colposcopic biopsy data by 31 March 2021. The University of Fukui Ethics Review Board approved this study (20200014) on 30 April 2020.

### 2.3. Statistical Analyses

Categorical variables are presented as frequencies and proportions. CIN2+ was defined as CIN2, CIN3, ADC in situ, SCC, or ADC. The prevalence of detected CIN2+ per 1000 screened and invited for HPV self-sampling with 95% confidence intervals (CI) was calculated [43]. We conducted all statistical analyses with Easy R (EZR) (version 1.42) (Saitama Medical Center, Jichi Medical University, Saitama, Japan), a graphical user interface for the R (The R Foundation for Statistical Computing, Vienna, Austria) [44].

## 3. Results

Figure 1 shows the demographic characteristics of the study participants. A total of 3489 women from the four municipalities who had not undergone screening for 5 years or more were included in this study. Overall, 10.6% (370/3489) of participants agreed to participate in the study and requested an HPV self-sampling kit. Furthermore, 77.3% (286/370) of participants returned the kit, accounting for an HPV testing rate of 8.2% (286/3489 of the eligible population). Moreover, 21.3% (61/286) of participants who returned the kit underwent cytology testing after returning the kit, and 18.0% (11/61) of the cytology tests were performed during pregnancy examinations. The positivity rate of HPV self-sampling was 13.7% (39/285). Moreover, 61.5% (24/39) of participants with a positive HPV self-sampling result underwent cytology testing. Thereafter, 15% (37/246) of participants with a negative HPV self-sampling result underwent cytology testing, and 8.4% (24/286) who returned the kit underwent cytology testing after a positive HPV result from self-sampling. Additionally, 14.3% (12/84) of participants who did not return the kit underwent cytology testing. The overall participation of cervical cancer screening after the agreement to HPV self-sampling consisted of participants who returned the HPV-self-sampling kit (*n* = 286) and those who did not return the kit but underwent cytology testing (*n* = 12). These were participants who received some form of screening after consenting to HPV self-sampling. The rate of participation of cervical cancer screening after the agreement to HPV self-sampling was 80.5% (298/370).

In terms of the HPV types found in HPV-positive participants (*n* = 39), there were 5 cases of HPV16, 1 case of HPV16 + Others, 1 case of HPV18, and 32 cases of HPV Others. HPV16/HPV18, which included HPV16 + Others, and other HPV types were observed in 17.9% (7/39) and 82.1% (32/39) of participants with positive HPV test results, respectively. Table 1 presents the results of HPV and cytology testing following HPV self-sampling. Overall, 13.0% (3/23) of sHPV-positive samples had low-grade cytology ratings, and 13.0% (3/23) of sHPV-positive samples had high-grade cytology ratings. Table 2 shows the results of colposcopic biopsy following cytology testing. Three cases of CIN2+ were identified, accounting for 4.9% (3/61) of participants who underwent cytology testing after returning the self-sampling kit. Among them, two cases of CIN3 were HPV16-positive. In the sHPV-negative samples, no cases of abnormal cytology and CIN2+ were reported. Table 3 shows the rates of CIN2+ detection in opt-in HPV self-sampling. CIN2+ was detected in 10.5 (3/286 × 1000) (95% confidence interval [CI], 2.1–31.8) per 1000 women who returned the HPV self-sampling kits and 0.9 (3/3489 × 1000) (95% CI, 0.2–2.7) per 1000 women who were invited for HPV self-sampling.

## 4. Discussion

In this study, we evaluated the compliance of opt-in HPV self-sampling using a screening framework in women in their 30s who have not undergone cervical cancer screening in the previous 5 years in Japan. To this end, we were able to select women who had not undergone cervical cancer screening and offered them to perform opt-in HPV self-sampling. The findings revealed that most of the unscreened women who participated in opt-in HPV self-sampling returned the sampling kit, with CIN2+ being detected in unscreened women who performed opt-in HPV self-sampling. However, approximately 40% of individuals with positive HPV self-sampling test results did not undergo cytology testing.

A noteworthy outcome in the present study was the high return rate of HPV self-sampling kits among women who had not undergone screening for >5 years and had participated in opt-in HPV self-sampling in Japan. Overall, 77.3% of participants returned the HPV self-sampling kits (HPV testing rate: 8.2% of the eligible population). Previous studies have reported that 79.5% and 63.2% of participants who were unscreened for 4 and 10 years, respectively, who agreed to opt-in HPV self-sampling returned the kit, with respective return rates of 20.7% and 8.15% [26,27]. In a report from Norway, the participation rates with reminders to attend regular screening, opt-in HPV self-sampling, and opt-out HPV self-sampling were 4.8%, 17.0%, and 27.7%, respectively [45]. In the present study, opt-in HPV self-sampling had a good return rate, as in other studies. Potentially, compared with that noted using an opt-out method, the non-use of the kit may be reduced using an opt-in method, and this may also reduce kit waste. The participation rate in this study was consistent with that in previous reports.

In the present study, despite the successful return of kits, a small number of individuals underwent cytology testing after a positive HPV self-sampling result. For the effective use of HPV self-sampling, a system that encourages subsequent cancer screenings should be implemented. A previous study showed that 86.5% of opt-in participants aged 30–64 years whose HPV self-sampling test results were positive underwent cytology testing [46]. Another study showed that 79.1% of opt-in participants aged 20–50 years whose HPV self-sampling test results were positive underwent cytology testing [47]. These differences among studies may be attributed to differences in study populations, particularly as the present study only included women in their 30s. The fundamental drawbacks of cytology might affect the cytology rate after HPV self-sampling. A lack of knowledge about HPV and cervical cancer might affect the triage cytology rate [48]. In this specific age group, women who return the kits need to be encouraged to undergo cytology testing.

Opt-in HPV self-sampling detected CIN2+ in unscreened Japanese women in their 30s. In the present study, CIN2+ was detected in 10.5 per 1000 women who returned the HPV self-sampling device and in 0.9 per 1000 women who were invited for HPV self-sampling. A previous study showed that CIN2+ was detected in 18.5 per 1000 people who returned the HPV self-sampling kit and in 1.5 per 1000 people who were invited for HPV self-sampling [49]. The number of CIN2+ cases was higher than that of CIN1 cases in this study. The study participants were unscreened, which is considered a high-risk CIN2+ population. In a previous study, CIN1 was detected in 28% of HPV-positive women after HPV self-sampling, whereas CIN2+ was detected in 32% [23]. The present study findings suggest that HPV self-sampling holds promise in detecting CIN2+ in unscreened Japanese women in their 30s.

The risk of contracting HPV is high for women who do not undergo cervical cancer screening. In the present study, the positivity rate for HPV self-sampling was 13.7%. A previous study has reported a 15.7% positivity rate for HPV self-sampling among unscreened women aged 30–39 years [49], indicating similar HPV positivity rates as that in the present study. We previously conducted a study of HPV physician-sampling in the same prefecture, and the HPV positivity rate of women in their 30s was 10.1% with HPV physician-sampling [11]. This study suggests a higher HPV positivity rate among unscreened women, indicating that this population has a high risk of HPV infection. Long-term hr HPV infection leads to cervical cancer [50]. To prevent cervical cancer, unscreened women should be encouraged to undergo cervical cancer screenings.

The communication method plays a crucial role in encouraging the participation of unscreened women in HPV self-sampling. As reported previously, the opt-out method has a higher participation rate than the opt-in method for cervical cancer screening [26]. Communication with unscreened women via electronic platforms might be essential to fill this gap between the opt-in and opt-out approaches. Lam et al. [51] reported the effectiveness of electronic communication platforms in inviting women who had not been screened for 4–6 years to participate in opt-in HPV self-sampling. They provided a mobile-friendly web page with multilingual instructions and video animations, along with the usual communication channels, such as letters or phone [51]. Among their participants, 31.7% requested a home test, and 20% returned the test kit to the laboratory. In another study, online communication methods were used more frequently by young women [52]. In the present study, unscreened women received a mail only to request the sampling device. A 2020 survey has revealed that 92.1% of individuals aged 30–39 years in Japan own a smartphone [53,54]. Therefore, electronic communication platforms might be promising to improve the participation rates in opt-in HPV self-sampling.

In an organized screening, interpreting the genotyping results from samples obtained via self-sampling is challenging. In a previous study, HPV16/HPV18 was detected as a type of CIN2+ hr HPV [9]. Only three cases of CIN2+ were detected in the present study. Three participants were HPV16-positive and underwent cytology testing. Moreover, two of the three HPV16-positive cases were CIN3, suggesting a high-risk association between HPV16 and CIN2+. However, HPV-based screening has a lower specificity than cytology-based screening, resulting in an increased number of colposcopies [53,54]. An unnecessary colposcopy may result in physical and psychological discomfort, as well as the overtreatment of degenerative cervical intraepithelial neoplasia [40]. In the screening framework in the Netherlands, women with an hr HPV-positive result on self-sampling are advised to undergo an assessment of the cytological smear taken by their general practitioner. Then, if the cytology result is ASCUS or worse, colposcopy is performed [13]. In Japan, follow-up methods should be developed to achieve high CIN2+ detection efficiency while mitigating the harmful influence of HPV genotyping results from self-sampling.

This study has some limitations. First, we investigated only unscreened women in their 30s and did not include unscreened women of other age groups. Second, this study was designed to evaluate the effects of introducing opt-in HPV self-sampling within the screening framework; therefore, a direct comparison with opt-out HPV self-sampling or reminder mail was not made. Third, data were collected in only few municipalities in Japan, indicating the need for larger, multicenter studies with follow-up to generalize and understand the short- and long-term impact of the opt-in method on HPV self-sampling. In the present study, participation in opt-in HPV self-sampling was low, and the kit returns rate was not high. Finally, participation in triage cytology for HPV-positive tests was low, resulting in a low detection rate of CIN2+. The use of HPV-related biomarkers such as DNA methylation on self-samples, which could detect CIN3, may be more effective than cytology triage [17,55]. After a positive result for HPV self-sampling, triage methods that do not rely on cytology may be beneficial.

## 5. Conclusions

In this study, the opt-in HPV self-sampling approach for unscreened women resulted in a notably low rate of self-sampling kit orders, with only 10.6% of the invited women opting to participate. While a relatively high return rate was noted for the kits that were ordered, the compliance for follow-up cytology among these women was low. Additionally, the inherent limitations in the sensitivity of cervical cytology further restricted the number of women referred for colposcopy. Consequently, out of the 3489 women initially invited for screening, CIN2+ was detected in only three cases, underscoring the need for more effective strategies for cervical cancer screening.

## Figures and Tables

**Figure 1 healthcare-12-00599-f001:**
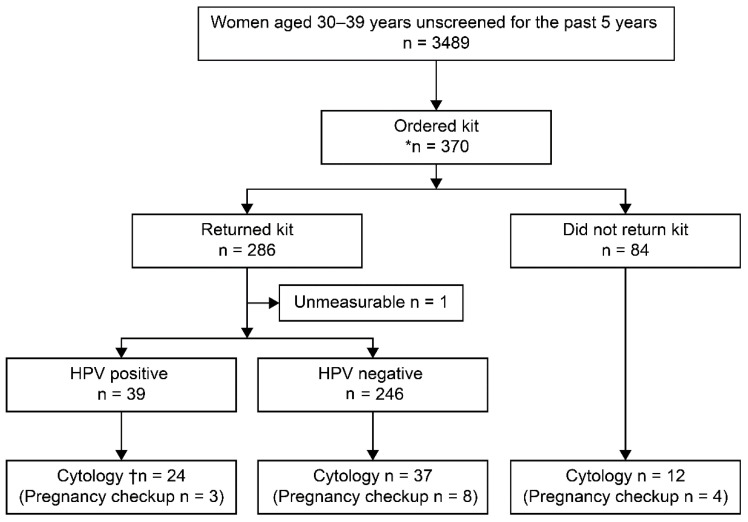
Study flowchart. HPV, human papillomavirus. * Initially, 379 participants agreed to participate, but 9 participants later withdrew their consent. ^†^ There was one unsatisfactory Pap test.

**Table 1 healthcare-12-00599-t001:** Association between HPV self-sampling and cytology results.

	Cytology Result		
	Normal (NILM)	Low-Grade (ASCUS/LSIL)	High-Grade (HSIL)
sHPV-positive, *n* = 23 *	17 (2) ^†^	3	3
sHPV-negative, *n* = 37	37 (8) ^†^	0	0

ASCUS, atypical squamous cells of undetermined significance; HPV, human papillomavirus; HSIL, high-grade squamous intraepithelial lesion; LSIL, low-grade squamous intraepithelial lesion; *n*: number of cases; NILM, negative for intraepithelial lesion or malignancy; sHPV, HPV self-sampling. * One case of sHPV-positive had the unsatisfactory cytology test in pregnancy checkup. ^†^ Numbers in parentheses present the cancer screening during the pregnancy checkup.

**Table 2 healthcare-12-00599-t002:** Pathology after HPV self-sampling.

Cases	sHPV Type	Cytology	Pathology
1	Others	ASCUS	Cervicitis
2	Others	LSIL	CIN2
3	16	HSIL	CIN3
4	16, others	HSIL	CIN3
5	Others	HSIL	CIN1

ASCUS, atypical squamous cells of undetermined significance; CIN, cervical intraepithelial neoplasia; HPV, human papillomavirus; HSIL, high-grade squamous intraepithelial lesion; LSIL, low-grade squamous intraepithelial lesion; sHPV, HPV self-sampling. Others: HPV-31, HPV-33, HPV-35, HPV-39, HPV-45, HPV-51, HPV-52, HPV-56, HPV-58, HPV-59, HPV-66, and HPV-68.

**Table 3 healthcare-12-00599-t003:** CIN2+ detection from HPV self-sampling.

	*CIN2+ per 1000 for Opt-In sHPV
sHPV screened (95% CI)	10.5 (2.1–31.8)
sHPV invited (95% CI)	0.9 (0.2–2.7)

CI, confidence interval; CIN, cervical intraepithelial neoplasia; HPV, human papillomavirus; sHPV, HPV self-sampling. *CIN2+ was defined as CIN2, CIN3, adenocarcinoma (ADC) in situ, squamous cell carcinoma, or ADC.

## Data Availability

The data presented in this study are available upon request from the corresponding authors.

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
