# Peer review of "Evaluating Opt-In Vaginal Human Papillomavirus Self-Sampling: Participation Rates and Detection of High-Grade Lesions (CIN2+) among Unscreened Japanese Women Aged 30–39"

_healthcare, 2024, doi:10.3390/healthcare12050599_

Round 1

Reviewer 1 Report

Comments and Suggestions for Authors

The authors evaluate the compliance of opt-in HPV self-sampling screening approach for women aged 30-39, resided in Echizen, Katsuyama, Ono City and Takahama Town in Fukui prefcture in Japan. I suggest the authors to enhance the content, method, presentation and interpretation of the results.

Vaccination and screening are the best strategy for cervical cancer prevention. What is the HPV vaccine coverage for women between 30-39 in Japan? What is the current screening guideline and strategy in Japan? What are the screening barriers? Has the op-in HPV self-sampling screening approach been implemented in Japan? The participation rate is only about 10% (370/3489=0.106 or 286/3489=0.08). How is the participation rate in this study compared to other approaches (i.e., reminder approach)?  Is this an acceptable rate?

The authors also need to clarify the screening strategies. “If the test result was HPV positive, participants were instructed to undergo organized cervical cancer screening by cytology” (Line 124-125). Why were those negative women going through cytology? Line 173, 3 out of 61 with cytology is CIN2+. This included those women who were HPV negative. How many were CIN2+ among those 39 women with HPV positive? What might be the reason that women tested positive did not go through cytology?

What is incidence of cervical cancer for women resided in Fukui prefecture compared to the other part of Japan? How is the access to healthcare compared to other parts of Japan? Can the results from this study be generalized to women in Japan? How is the CIN2+ detection rate compared with other screening approach? Is this opt-in screening approach a cost-effective approach?  

The authors need to enhance the method including statistical analysis, better link the method and results; also enhance the clarity of the tables.

Author Response

Dear Reviewer 1,

I apologize for delivering the review response so close to the deadline. Enclosed, you will find the response in a Word file. Thank you for your attention to this matter.

Kind regards,

Taro Ito

Reviewer 2 Report

Comments and Suggestions for Authors

Title: Consider adding “Vaginal” Self Sampling (SS is also available for Urine & menstrual period material)

This study has been conducted in Japanese prefectures, by a group with previous experience in the field. In the introduction section the author’s should provide more insight on SS aspects, as well as references to review articles elucidating issues of vaginal SS strategies (e.g. among others, doi: 10.3390/cancers15061669).

Despite meticulously prepared/conducted, this study is hampered by some drawbacks:

1. Low opt in/participation rate.

2. Low kit Return Rate (>20% of participants not returning the kit they ordered themselves is unacceptably high) - (return rate was 8.2% (286/3489) of the eligible population -line 151).

Collectively points 1 & 2 result in low numbers of CIN2+ harvest (only three individuals)

3. Suboptimal follow up rates of positive self samples in the community “However approximately 40% of individuals whose HPV self-sampling test results were positive did not undergo cytology testing” (lines 203-204). The authors do acknowledge that “To utilize HPV self-sampling, it is vital to implement a system that encourages subsequent cancer screenings” (lines 216-7).

Based on the study’s design, the authors selected to triage positive HPV DNA SS results by cytology in the community. Enhancing efficiency & cost-effectiveness in SS triage is better achieved by assessing other HPV-related biomarkers in THE SAME SS MATERIAL (currently, predominantly by methylation tests), therefore skipping the need for a new triage appointment with a physician. Furthermore, cytology’s fundamental drawbacks also affect its use a SS triage test. Effective triage strategies in SS interventions are of utmost importance as continuity of care is otherwise breeched; besides medicolegal issues might prospectively arise.

4. Cost-effectiveness considerations: Only 3 CIN2+ cases have been harvested in this intervention.

Authors are encouraged to address the aforementioned points in the Discussion section (e.g. line 274 – limitations of the study), without embarking upon a major revision.

The manuscript would also benefit from language polishing: e.g. “A previous study has shown that HPV16/HPV18 is a CIN2+ hr HPV type” (lines 203-204).

Comments on the Quality of English Language

Language Polishing is mandatory

Author Response

Dear Reviewer 2,

I apologize for delivering the review response so close to the deadline. Enclosed, you will find the response in a Word file. Thank you for your attention to this matter.

Kind regards

Taro Ito

Reviewer 3 Report

Comments and Suggestions for Authors

Taro et al. have conducted a study to introduce human papillomavirus (HPV) self-sampling in Japan, specifically targeting women aged 30-39 who require preconception care and have not been screened for cervical cancer. The research evaluated the impact of opt-in HPV self-sampling in this demographic. Participants used the Evalyn® Brush for self-sampling and HPV testing was performed using the Cobas 4800 system, with results mailed to them. Out of 3489 eligible, unscreened women from four municipalities in Fukui prefecture, 10.6% requested the self-sampling kit, and 77.3% of these returned it. The overall return rate was 8.2%, with a 13.7% positivity rate for HPV. Among the positive cases, 61.5% underwent cytology testing, leading to the detection of three cases (10.5 per 1000 women) of cervical squamous intraepithelial neoplasia 2 or worse (CIN2+). The study highlights that opt-in HPV self-sampling is effective in detecting CIN2+ in unscreened women, but also notes the need for increased participation in cytology testing among those who return the kit.

Comments

Research has consistently shown that opt-out strategies yield significantly higher compliance rates compared to opt-in strategies, primarily due to the inherent human tendency to stick with default options. The study employs cytology as a secondary screening (triage) for women with positive HPV self-sampling results. While this adds an extra diagnostic step, it is crucial to note that cytology's relatively lower sensitivity could potentially delay the detection of high-grade lesions. Considering cytology's limited sensitivity, a more proactive approach might be the direct referral of HPV DNA-positive women for colposcopy and biopsy, especially for those who have not been screened in over a decade. This could significantly streamline the diagnostic process. Such a streamlined approach not only reduces the frequency of call-backs and follow-ups but also accelerates the identification and treatment of CIN2+ lesions. This is particularly important in populations with historically low screening rates, where simplifying the screening process could enhance overall compliance and healthcare outcomes.

While the study by Taro et al. provides valuable insights into opt-in HPV self-sampling compliance, a critical aspect that warrants further discussion is the initial participation rate. The study reports that only 10.6% (370 out of 3489) of the invited women actually ordered the self-sampling kit. This low response rate represents a significant limitation, especially when considering the potential outcomes of a more inclusive approach.

Had the self-sampling kits been sent directly to all 3489 women, as per an opt-out strategy, the response rate might have been considerably higher. For instance, even with a conservative estimate of a 30% return rate, this would have resulted in over 1,000 samples for HPV testing, rather than the 370 obtained. With the observed positivity rate of 13.7% in HPV tests, this approach could have led to more than 137 women being identified for follow-up, as opposed to 39.

Furthermore, if all these 137 women were referred directly for colposcopy and biopsy, the number of confirmed CIN2+ cases might have been higher than the three cases identified through the cytology triage. This alternative strategy could potentially have a more significant impact on detecting and managing CIN2+ cases. Therefore, while the study highlights important aspects of compliance with opt-in self-sampling, the overall effectiveness of the screening program could be substantially improved by reconsidering the initial approach to kit distribution.

Aasbø G, Tropè A, Nygård M, Christiansen IK, Baasland I, Iversen GA, Munk AC, Christiansen MH, Presthus GK, Undem K, Bjørge T, Castle PE, Hansen BT. HPV self-sampling among long-term non-attenders to cervical cancer screening in Norway: a pragmatic randomised controlled trial. Br J Cancer. 2022 Nov;127(10):1816-1826. doi: 10.1038/s41416-022-01954-9. Epub 2022 Aug 23. PMID: 35995936; PMCID: PMC9643532.

Enerly E, Bonde J, Schee K, Pedersen H, Lönnberg S, Nygård M. Self-Sampling for Human Papillomavirus Testing among Non-Attenders Increases Attendance to the Norwegian Cervical Cancer Screening Programme. PLoS One. 2016 Apr 13;11(4):e0151978. doi: 10.1371/journal.pone.0151978. PMID: 27073929; PMCID: PMC4830596.

Minor revisions

Line 2-4, "Evaluating Opt-in HPV Self-Sampling: Participation Rates and Detection of High-Grade Lesions (CIN2+) Among Unscreened Japanese Women Aged 30–39"

Line 15-28, "Abstract: This study aimed to assess the implementation of opt-in human papillomavirus (HPV) self-sampling among Japanese women aged 30–39 years who had not undergone cervical cancer screening, particularly focusing on those requiring preconception care. The study evaluated both the response to the opt-in approach and the effectiveness in detecting cervical squamous intraepithelial neoplasia 2 or worse (CIN2+). Participants used the Evalyn® Brush for self-sampling, with HPV testing conducted using the Cobas 4800 system and results communicated via mail. Out of 3,489 eligible, unscreened women from four municipalities in Fukui prefecture, only 10.6% (370/3489) requested the self-sampling kit. Of these, 77.3% (286/370) returned the kit, equating to an 8.2% return rate from the eligible population. The HPV positivity rate among the samples was 13.7% (39/285), yet only 61.5% (24/39) of those with positive HPV results proceeded to cytology testing. Subsequently, three cases of CIN2+ were detected (10.5/1000). While the study demonstrates a reasonable kit return rate and the capability of opt-in HPV self-sampling to detect CIN2+ cases in unscreened women, the low ordering rate of the kits and suboptimal compliance for follow-up cytology testing highlight significant challenges. The findings suggest the need for more effective strategies to enhance participation in cervical cancer screening programs."

Line 33-46, "The incidence of cervical cancer among women of reproductive age is on the rise in Japan. Data from the 2020 Global Cancer Observatory reveals that the incidence rate of cervical cancer in Japan for the age group of 20 to 44 years is 27.7 per 100,000. This rate is notably higher compared to Australia (10 per 100,000) and the United States (9 per 100,000) [1]. Given this concerning trend, it is crucial for Japan to implement urgent measures to reduce the incidence of cervical cancer during reproductive age. One significant contributing factor to the high incidence and mortality rates of this disease in Japan is the low screening rate [1,2]. Japan's cervical cancer screening rate stands at 42.1%, markedly lower than that in the United States (84.5%) and the United Kingdom (78.1%) [3]. This underscores the need to encourage women, especially those who have not participated in screening programs, to undergo cervical cancer screening. Additionally, the average age of first childbirth in Japan has risen from 28 years in 2000 to 30.7 years in 2019 [4], a demographic shift that aligns with the finding that most advanced cervical cancers occur in women who have not been screened [5]. Therefore, as part of routine preconception care, it is of paramount importance to increase cervical cancer screening among women aged 30–39 years [6]."

Line 47-55, "Persistent infections with human papillomaviruses (HPV) are a recognized cause of cervical cancer [7]. HPV tests are known for their higher sensitivity in detecting cervical squamous intraepithelial neoplasia 2 or worse (CIN2+) compared to cytology testing alone [7–9]. The process of HPV self-sampling for cervical cancer detection involves participants collecting a vaginal sample themselves and sending it for HPV testing [10,11]. The concordance rate between HPV self-sampling and physician-collected sampling lies between 85–90%. Moreover, the sensitivity of HPV self-sampling for detecting CIN2+ is comparable to that of samples collected by physicians [10,12,13]. Importantly, self-sampling has been shown to be an effective strategy in motivating women who have not previously participated in cervical cancer screening. Its effectiveness surpasses that of conventional mail-out invitations [14–19]."

Line 56-68, "Unscreened women can be provided with HPV self-sampling kits through two approaches: opt-in or opt-out. The opt-out method involves sending HPV self-sampling kits to unscreened women without requiring their prior consent, whereas the opt-in method sends kits only after confirming the women's willingness to receive them. Studies have shown that the opt-out approach is more effective in encouraging participation of unscreened women in cervical cancer screening compared to the opt-in method [14]. However, the opt-out strategy has its drawbacks: it is costlier and less environmentally sustainable. For instance, in opt-in methods, about 10–20% of the kits are not utilized, but this figure rises to approximately 70–80% in opt-out methods, as evidenced in studies from Sweden and Slovenia [19,20]. Consequently, the kit return rate, which serves as an indicator of compliance, is higher with the opt-in approach [19,20]. The Netherlands has adopted opt-in HPV self-sampling as a strategy to encourage unscreened women to participate in cervical cancer screening [11]. However, the compliance of unscreened women in Japan towards the opt-in method remains unexplored."

Line 69-73, "This study was designed to assess the compliance with opt-in HPV self-sampling within the framework of actual cervical cancer screening among unscreened Japanese women aged 30–39 years. Our objectives were twofold: firstly, to ascertain the response rate to opt-in HPV self-sampling among these unscreened women; and secondly, to determine the detection rate of cervical squamous intraepithelial neoplasia 2 or worse (CIN2+) in this group through the use of opt-in HPV self-sampling."

Table 1 requires modification due to the presence of numerous cells containing only the numbers 0 or 1. To simplify, please consolidate the HPV results into two categories: HPV positive and HPV negative. Similarly, categorize cytology results into three groups: Normal (NILM), low-grade (ASC-US / LSIL), and high-grade (HSIL). This will result in a more concise 2 x 3 table, instead of the current 7 x 6 table which has many cells with minimal content. Additionally, data on women with unsatisfactory Pap tests can be included as a footnote to the table for clarity.

Line 283-285, "In conclusion, the opt-in HPV self-sampling approach for unscreened women resulted in a notably low rate of self-sampling kit orders, with only 10.6% of the invited women opting to participate. While there was a relatively high return rate of the kits that were ordered, the compliance for follow-up cytology among these women was disappointingly low. Additionally, the inherent limitations in the sensitivity of cervical cytology further restricted the number of women referred for colposcopy. Consequently, out of the 3,489 women initially invited for screening, only 3 cases of CIN2+ were detected, underscoring the need for more effective strategies in cervical cancer screening."

Comments on the Quality of English Language

As a non-native English speaker, I have suggested some reformulations to improve the language in the manuscript. However, I recommend considering a review by a native English speaker to further enhance sentence construction, readability, coherence, and accuracy, ensuring a polished final presentation.

Author Response

Dear Reviewer 3,

I apologize for delivering the review response so close to the deadline. Enclosed, you will find the response in a Word file. Thank you for your attention to this matter.

Kind regards

Taro Ito

Round 2

Reviewer 1 Report

Comments and Suggestions for Authors

The authors evaluate the compliance of opt-in HPV self-sampling screening approach for women aged 30-39, resided in Echizen, Katsuyama, Ono City and Takahama Town in Fukui prefecture in Japan. I suggest the authors to better clarifying the outcome measures, the analysis approaches (e.g., How the frequency would be calculated? What was the denominator? How was the CIN2+ detection rate was calculated?) in the method section, better align the objects, analysis methods, results, tables and discussion. Other suggestions to consider:

Line 89-90, comparing total cost from two testing approaches does not seem to have much meaning. Cost per test conducted should be more meaningful besides comparing cost-effectiveness.

Line 158-161, were either positive or negative participants advised to go through cytology cancer screening? Was the cytology screening the same?

Line 181, there are no demographic characteristics data in Figure 1. It is helpful to have the demographic data if available.

Line 186, The return rate should be based on participants who received kits. For those people who did not receive kits, it was not possible for them to return kits.   

Line 197, do you mean the overall participation rate of cytology?

Line 198-200, why exclude 4 pregnant women who did not return kit but not those pregnant women who were HPV negative?

Line 205, Table 1, based on Figure 1, there were 24 HPV positive went through cytology, why only 23 listed in Table 1?

Line 205-210, these numbers were not in Table 1.

Line 212, based on Table 2, there were only 2 CIN2+; there were more than 3 cases of CIN3

Line 216, how 10.5 samples were obtained? How were detection rates calculated? Please clarify in the method section. 

Author Response

Dear Reviewer,

Apologies for the delayed response, but please find attached the answers to your questions in a Word document. We kindly ask for your review. Thank you.

Kindl regards, Taro Ito
